# Effects of Late Conversion from Twice-Daily to Once-Daily Slow Release Tacrolimus on the Insulin Resistance Indexes in Kidney Transplant Patients

Valeria Cademartori [1], Fabio Massarino [1], Emanuele L. Parodi [1], Ernesto Paoletti [1], Rodolfo Russo [1], Antonella Sofia [1], Iris Fontana [2], Francesca Viazzi [1,3], Pasquale Esposito [1,3,*] and Giacomo Garibotto [3]

1   Clinica Nefrologica, Dialisi e Trapianto, IRCCS Ospedale Policlinico San Martino, 16132 Genova, Italy; valeriacademartori@alice.it (V.C.); fabio.massarino@hsanmartino.it (F.M.); emanuele.l.parodi@gmail.com (E.L.P.); ernesto.paoletti@hsanmartino.it (E.P.); rodolfo.russo@hsanmartino.it (R.R.); antonella.sofia@hsanmartino.it (A.S.); Francesca.Chiara.Viazzi@unige.it (F.V.)
2   UOS Chirurgia del Trapianto renale, IRCCS Ospedale Policlinico San Martino, 16132 Genova, Italy; iris.fontana@hsanmartino.it
3   Nephrology Division, Department of Internal Medicine, University of Genoa, Viale Benedetto XV, 6, 16132 Genoa, Italy; gari@unige.it
*   Correspondence: pasquale.esposito@unige.it

**Abstract:** The use of tacrolimus (Tac) may be involved in the development of new-onset diabetes after transplantation (NODAT) in a dose-related manner. This study aimed to evaluate the effects of a standard twice-daily formulation of Tac (TacBID) vs. the once-daily slow-release formulation (TacOD) on the basal insulin resistance indexes (Homa and McAuley), and related metabolic parameters, in a cohort of kidney transplant patients. We retrospectively evaluated 20 stable renal transplant recipients who were switched from TacBID to TacOD. Blood levels of Tac were analyzed at one-month intervals from 6 months before to 8 months after conversion. Moreover, Homa and McAuley indexes, C-peptide, insulin, HbA1c, uric acid, triglycerides, low-density lipoprotein (LDL) and high-density lipoprotein (HDL)-cholesterol serum levels and their associations with Tac levels were evaluated. We observed a significant decrease in Tac exposure ($8.5 \pm 2$ ng/mL, CV 0.23 vs. $6.1 \pm 1.9$ ng/mL, CV 0.31, TacBID vs. TacOD periods, $p < 0.001$) and no significant changes in Homa ($1.42 \pm 0.4$ vs. $1.8 \pm 0.7$, $p > 0.05$) and McAuley indexes ($7.12 \pm 1$ vs. $7.58 \pm 1.4$, $p > 0.05$). Similarly, blood levels of glucose, insulin, HbA1c, lipids, and uric acid were unchanged between the two periods, while C-peptide resulted significantly lower after conversion to TacOD. These data suggest that in kidney transplant recipients, reduced Tac exposure has no significant effects on basal insulin sensitivity indexes and metabolic parameters.

**Keywords:** kidney transplant; once-daily tacrolimus; insulin resistance; HOMA index; McAuley index; NODAT

## 1. Introduction

New-onset diabetes after transplantation (NODAT) is a serious complication of transplantation associated with increased mortality and morbidity, and high rates of cardiovascular disease and infection, which are the leading causes of death in renal transplant recipients. [1]. The NODAT incidence varies across different studies, ranging from 0.4% to 50%. In particular, incidences between 7%–30% in the first year after transplantation have been reported [2,3]. NODAT recognizes multiple pathogenic mechanisms resulting in an imbalance between insulin production and target tissue insulin demand. In the transplant setting, this disparity may occur as a result of insulin resistance, increased insulin metabolism, or diminished insulin secretion [4,5]. So, for example, the increased prevalence

in the first few months after kidney transplantation may be justified by the superimposition of transplant-specific factors (e.g., weight gain and diabetogenic immunosuppressive therapies) on the baseline metabolic milieu of predisposed individuals [6]. The type of immunosuppression may account for around 70% of the variability in the incidence of NODAT [7,8]. So, steroids increase insulin resistance by increasing hepatic neoglucogenesis and inhibiting peripheral glucose uptake [9].

The calcineurin inhibitors (CNIs), cyclosporine (CsA) and tacrolimus (Tac), represent the cornerstone of the immunosuppressive therapy for renal transplantation, and their role in the pathogenesis of NODAT is well-documented [10].

CsA and Tac share a similar therapeutic action, and common mechanisms may explain their diabetogenic potential. These include decreased insulin secretion, increased insulin resistance, and inhibition of steroid metabolism. Both CNIs cause reversible toxicity to islet cells and may directly affect the transcriptional regulation of insulin expression [11].

However, the use of tacrolimus has shown a higher association than cyclosporine for abnormal glucose metabolism, causing more severe swelling and vacuolization of islet cells [12,13].

Previous studies analyzed the role and the cellular localization of the specific binding proteins for both Tac (FKBP-12 bp) and CsA (cyclophilin) to justify the toxicity of Tac on islet cells.

FKBP-12 bp is highly concentrated in the insulin-producing beta-cells of the pancreas, whereas cyclophilin appears preferentially localized into the heart, liver, and kidney [14,15].

Accordingly, it was observed that Tac reduces the number of endocrine secretory granules in human pancreatic beta-cells [5]. In renal transplant patients, it has been shown that Tac decreases insulin secretion in a dose-related manner, and blood levels >15 ng/mL during the first month after transplantation represents a relevant risk factor for NODAT [16]. Furthermore, calcineurin signaling may be involved in modulating insulin sensitivity in skeletal muscle. Various signaling pathways are responsible for the remodeling of skeletal muscle [17–20]. It has been demonstrated that the inhibition of the calcineurin/Nuclear factor of activated T-cells (NFAT) pathway in skeletal muscle could promote the transcription of insulin-resistant myosin fast fibers contributing to the development of insulin resistance [21].

Insulin resistance (IR) can be assessed by standard methods, like the euglycemic insulin clamp method, intravenous glucose tolerance test, and minimal model approximation of the metabolism of glucose. In addition to these standard methods, there are indirect methods for the assessment of IR: homeostasis model assessments (HOMA-IR) and McAuley index (McA) [22,23]. More recently, beyond the standard twice-daily formulation of Tac (TacBID), a new once-daily slow-release formulation (TacOD) has been introduced in the clinical practice. The use of this formulation seems to be associated with a reduced variability of the drug trough level and a low incidence NODAT when used in de novo kidney transplant patients [24].

This study aimed to evaluate the effects of the conversion from TacBID to TacOD on the basal insulin resistance indexes (HOMA and McAuley) and related metabolic parameters in a cohort of kidney transplant patients.

## 2. Materials and Methods

### 2.1. Study Design

We reviewed data from non-diabetic renal transplant recipients followed at the Nephrology, Dialysis and Transplantation Division, University of Genoa, who switched from standard twice-daily formulation of Tac to the once-daily slow-release formulation. All patients were on a double immunosuppressant regimen (Tac/Mycophenolate mofetil-MMF) and were steroid-free.

Other drugs, including antihypertensive drugs, sodium bicarbonate, calcium carbonate, calcitriol, cinacalcet, and erythropoietin were prescribed as appropriate for each patient.

We evaluated a cohort of patients receiving a kidney graft for more than one year, with stable kidney function (proteinuria < 500 mg/day and serum creatinine fluctuation < 20%). Inclusion criteria were: age > 18 years, and no rejection in the previous 12 months. Exclusion criteria were: combined pancreas/kidney transplantation, history of diabetes mellitus before transplantation, history of NODAT, combined pancreas/kidney transplantation, pregnancy and current or past history of malignancy.

The patients were converted to TacOD on a 1:1 mg basis of the daily dose. If needed, subsequent dose adjustments were allowed during the next 2 weeks to stay within the center trough concentration target for Tac (5–10 ng/mL).

Blood levels of Tac were analyzed at one-month intervals from 6 months before to 8 months after conversion. Homa and Mc Auley indexes, C- peptide, insulin, glycated hemoglobin (HbA1c), triglycerides (TG), LDL, HDL-cholesterol, and uric acid serum levels and their associations with blood levels of Tac were also evaluated.

This study was performed in accordance with the Declaration of Helsinki and it was approved by the Local Ethical Committee. Informed consent was obtained from all participants.

### 2.2. Clinical and Laboratory Parameter Analysis

The patients met in the morning after a 12-h overnight fast; each participant's weight, height, and blood pressure were measured and recorded. The blood sample for the measurement of serum insulin and C-peptide was drawn in vacutainer tubes without additives and left at room temperature and storage at –20 °C until analyzed. Ordinary clinical chemistry and hematology parameters included creatinine, HbA1c, TG, LDL, HDL cholesterol, and acid uric were also analyzed for in other samples. Blood samples for the measurement of Tac whole blood concentrations were drawn in EDTA vacutainer tubes and stored at—20 °C until the analyses were performed. Samples were aimed to be drawn before administration of the morning dose of TAC.

Serum insulin and C-peptide were analyzed by ELISA-kits using a microplate reader. The intra and inter-assay coefficients of variation were <10% for the assays. Whole-blood Tac concentrations were analyzed using the Architect Tacrolimus Assay on An Architect i2000SR system (Abbott Diagnostics, Abbott Park, IL, USA); other clinical chemistry and hematology analyses were performed at the hospital laboratory with standard kits.

Insulin resistance was assessed by indirect methods (Homa and McAuley indexes) using the equations mentioned below [22,23]:

- McAuley (McA) = exp [2.63 − 0.28 ln (insulin in mU/L) − 0.31 ln (triglycerides in mmol/L)]
- HOMA = fasting Glucose(mg/dl) × fasting Insulin(μU/mL) / 405.

Patients were considered as insulin resistant when McA ≤ 5.8 and HOMA ≥ 2.

Estimated glomerular filtration rate (eGFR) was calculated by using the Chronic Kidney Disease Epidemiology Collaboration (CKD-EPI) equation [24].

### 2.3. Statistical Analysis

Data are presented as mean ± standard deviation (SD) unless otherwise specified. Paired student *t*-test or nonparametric tests, were used to assess the differences between patients in different treatment periods. Variability of tacrolimus trough levels was assessed using the coefficient of variation (CV), calculated as follows: CV (%) = (SD/mean tacrolimus concentration) × 100 [25].

Pearson's test was used to assess the correlations between Tac trough levels and laboratory variables. All statistical analyses were performed using Stata software (Stata 13.1, Stata Corporation, College Station, TX, USA). A 2-tailed *p* value < 0.05 was considered statistically significant.

## 3. Results

### 3.1. Patient Characteristics

Twenty kidney transplant patients who underwent conversion from Tac BID to TacOD were retrospectively studied. The median age was 52 years (range 26–79; 8 males/12 females).

Three had received a graft from a living donor. All the patients were on a double immunosuppressant regimen with mycophenolate mofetil, the dosage of which did not change during the evaluated study period. Patient characteristics are reassumed in Table 1.

**Table 1.** Patient characteristics at the basal evaluation, before the conversion from standard twice-daily formulation of tacrolimus (TacBID) to the new once-daily slow-release formulation of tacrolimus (TacOD).

|  | Total Population |
|---|---|
| N | 20 |
| Gender (M/F) | 8/12 |
| Age (years—IQR) | 52 (26–79) |
| BMI (kg/m$^2$) | 23.6 ± 3.6 |
| Time from transplantation (years) * | 3.2 ± 1.4 |
| Living donor transplantation, n (%) | 3 (15%) |
| Primary cause of ESRD, n (%) |  |
| - HTN/vascular | 7 (35) |
| - GLN | 8 (40) |
| - Hereditary | 3 (15) |
| Other/UN | 2 (10) |

Abbreviations: TacBID = twice-daily tacrolimus formulation, TacOD = once-daily slow-release tacrolimus formulation, BMI = body mass index, ESRD = end-stage renal disease; HTN = hypertension, GLN = glomerulonephritis, UN = unknown. * Time from transplantation corresponded to the duration of exposure to TacBID.

### 3.2. Effects of the Conversion from TacBID to TacOD

Conversion from TacBID to TacOD was scheduled on a 1:1 daily dose basis, with subsequent dose adjustments to stay within center trough concentration target for Tac (5–10 ng/mL).

Mean transplant age at the time of conversion was 3.2 ± 1.4 years. Notably, since each patient commenced TacBID treatment immediately after the transplant, transplant age at the conversion corresponded to the duration of exposure to TacBID.

The conversion was uneventful and no adverse events were reported during the study, including transplant rejection. As shown in Table 2, there were no significant differences in blood pressure, BMI, estimated glomerular filtration rate (eGFR), and other clinical chemistry after conversion.

**Table 2.** Clinical and biochemical data before and after the switch from TacBID to TacOD.

| N = 20 | TacBID | TacOD |
|---|---|---|
| Systolic blood pressure (mmHg) | 132 ± 12 | 130 ± 12 |
| Diastolic blood pressure (mmHg) | 78 ± 6 | 76 ± 5 |
| BMI (kg/m$^2$) | 23.6 ± 3.6 | 24.9 ± 3.4 |
| eGFR (ml/min./1.73 m$^2$) | 80 ± 1.2 | 80 ± 1.3 |
| Serum creatinine (mg/dL) | 1.2 ± 0.31 | 1.2 ± 0.39 |
| Total cholesterol (mg/dL) | 200 ± 28 | 201 ± 30 |
| HDL cholesterol (mg/dL) | 55 ± 0.4 | 56 ± 0.3 |
| LDL cholesterol (mg/dL) | 94 ± 0.5 | 95 ± 0.6 |
| TG (mg/dL) | 137 ± 48 | 113 ± 52 |

Abbreviations: TacBID = twice-daily tacrolimus formulation, TacOD = once-daily slow-release tacrolimus formulation, BMI = body mass index, eGFR= estimated glomerular filtration rate, HDL = high-density lipoprotein cholesterol, LDL = low-density lipoprotein cholesterol, TG = triglycerides.

After the switch to TacOD we observed a significant decrease in Tac exposure with a stable variability of Tac concentrations (8.5 ± 2 ng/mL, CV 0.23% vs. 6.1 ± 1.9 ng/mL, CV

0.31%, TacBID vs. TacOD, $p < 0.001$), but we did not observe significant changes in Homa ($1.42 \pm 0.4$ vs. $1.8 \pm 0.7$, TacBID vs. TacOD, $p = 0.07$) and McAuley indexes ($7.12 \pm 1$ vs. $7.58 \pm 1.4$, TacBID vs. TacOD, $p = 0.27$).

Evaluating metabolic parameters, we found no significant differences after the switch from TacBID to TacOD, except for C-peptide that resulted reduced after the conversion to TacOD (Table 3).

**Table 3.** Tac through levels. insulin resistance indexes and metabolic parameters before and after the switch from TacBID to TacOD.

| N = 20 | TacBID | TacOD | *p*-Values |
|---|---|---|---|
| Tac trough levels (ng/mL) | $8.5 \pm 2$ | $6.1 \pm 1.9$ | <0.001 |
| HOMA-index | $1.42 \pm 0.4$ | $1.8 \pm 0.7$ | 0.07 |
| McAuley-index | $7.12 \pm 1$ | $7.58 \pm 1.4$ | 0.27 |
| Glucose (mg/dL) | $93.4 \pm 19$ | $93.7 \pm 17$ | 0.6 |
| Insulin (mU/L) | $7.6 \pm 2.69$ | $8.8 \pm 3.8$ | 0.28 |
| C-peptide | $3.02 \pm 1.13$ | $2.5 \pm 0.8$ | <0.01 |
| HbA1c (%) | $5.64 \pm 0.6$ | $5.61 \pm 0.5$ | 0.48 |
| Uric Acid (mg/dL) | $5.36 \pm 1.17$ | $5.45 \pm 1.22$ | 0.7 |

Abbreviations: TacBID = twice-daily tacrolimus formulation, TacOD = once-daily slow-release tacrolimus formulation, HOMA = homeostatic model assessment, HbA1c = glycated hemoglobin.

Finally, we did not observe any association between Tac levels and blood levels of glucose, insulin, C-peptide, HbA1c, and uric acid (see Table 4).

**Table 4.** Correlations among Tac levels and metabolic parameters in clinically stable kidney transplant patients.

| Tac trough Levels N = 20 | Correlation Coefficient | *p*-Values |
|---|---|---|
| Glucose | 0.12 | 0.2 |
| Insulin | 0.03 | 0.8 |
| C-peptide | 0.2 | 0.06 |
| HbA1c | 0.16 | 0.1 |
| Uric Acid | $-0.2$ | 0.4 |

Abbreviations: TacBID = twice-daily tacrolimus formulation, TacOD= once-daily slow-release tacrolimus formulation, HOMA= Homeostatic model assessment, HbA1c= glycated hemoglobin.

## 4. Discussion

In our study, we found that late conversion from TacBID to TacOD led to a significant reduction in Tac trough levels, considered as the expression of a lower Tac exposure. However, the reduced Tac exposure was non accompanied by a parallel improvement in insulin sensitivity. This could be partially explained by the fact that the reduction in Tac trough level, even though statistically significant, were too small to determine a clinically measurable effect on glucose metabolism, also considering that at the time of the study, the center Tac target trough concentrations were 5–10 ng/mL (i.e., lower compared with Tac trough level of a few years ago). The chronic exposure to Tac could also explain these results; in fact, our patients were switched to TacOD after receiving TacBID for a long time. So, it is conceivable that previous chronic exposure to high Tac levels could have had a stable negative impact on glucose-stimulated insulin release. Other studies explored the short-time effects on glucose metabolism of conversion from standard Tac formulation to the slow release one. Midtvedt et al. evaluated hyperglycaemic clamp on TacBID and 4–6 weeks after switching to TacOD in 20 stable non-diabetic renal transplant recipients [26]. They found that, despite reduced Tac exposure, no change in insulin release or sensitivity occurred. Instead, Ruangkanchanasetr et al. evaluated the effect of conversion to TacOD in 28 patients, measuring HOMA index 4, 8, and 16 weeks after the switch [27]. Interestingly, they found that conversion to TacOD was effective in improving insulin resistance only in the subgroup of patients undergoing conversion within four years after the transplant.

So, these data corroborate our findings, obtained from a longer observation period, that a late switch between the two formulations of Tac did not have an effect on glucose metabolism.

Moreover, we observed a significant decrease of C-peptide levels with TacOD compared with TacBID. C-peptide is almost completely cleared by the kidney, but in our patients, renal function did not change significantly after the switch; so, improvement in renal function cannot be the reason for the reduction of C-peptide.

Historically, C-peptide, which is a part of the pro-insulin molecule and is secreted in equimolar amounts to insulin, has been used as a marker of insulin secretion in the differential diagnosis of diabetes [28]. However, significant positive correlations among C-peptide levels and metabolic parameters and markers of β cell function were also found, suggesting that the assessment of C-peptide levels may be a surrogate of insulin resistance [29]. Consequently, the decrease of C-peptide levels we observed after the switch from TacBID to TacOD may be the consequence of a reduced insulin requirement, following a partial improvement in insulin resistance.

However, this data is not in line with the trend of more specific insulin resistance indexes, and probably indicates that insulin secretion mechanisms should be more deeply analyzed. Finally, another critical point to consider as a consequence of insulin resistance after kidney transplant is the post-transplant weight gain [30,31]. Indeed, persistent NODAT may be associated with post-transplant weight gain, which may reflect the use of immunosuppressant medications after transplantation [32].

This is a crucial aspect, since a weight gain of ≥20% in the first post-transplant year is associated with poor survival outcomes [33]. In our study, conversion from TacBID to TacOD was not associated with a reduction in BMI, which, conversely, increased (with a weight gain of 9%), although in a not statistically significant manner, thus further confirming the absence of metabolic effects of late Tac conversion. Our study presents some limitations, including the retrospective design and the lack of specific analysis of insulin secretion and direct insulin resistance measurements. Moreover, the small number of patients did not allow for the performance of adequate multivariate analysis to assess the possible interconnections among Tac exposure, basal insulin resistance indexes, and related metabolic parameters. In this regard, a larger prospective interventional approach, comparing patients continuing TacBID vs. patients switched to TacOD, could be more informative about the metabolic effects of the conversion.

Finally, it should be considered that the gene profile of specific proteins, such as cytochrome P450 (CYP), may influence the pharmacokinetics of immunosuppressive drugs [34]. In particular, it has been found that genetic polymorphism of CYP3A5 affects Tac exposure after the conversion to TacOD [35].

Therefore, the assessment of CYP alleles could be meaningful in evaluating the effects of different Tac formulations. Further studies should be designed to investigate this intriguing aspect.

The above-reported considerations underline that our results, such as those of previous studies investigating the relationship between Tac exposure and insulin resistance, are not conclusive and many questions remain to be analyzed.

Nevertheless, our data suggest that in stable kidney transplant patients, the decrease of tacrolimus trough levels, within the therapeutic ranges, has no effects on basal insulin sensitivity indexes, and metabolic parameters.

**Author Contributions:** Conceptualization, V.C. and F.M.; Methodology, E.L.P.; Formal Analysis, E.P. and G.G.; Investigation, R.R., A.S. and I.F.; Writing F.V. and P.E.; Original Draft Preparation V.C. and P.E., Review & Editing, G.G. All authors have read and agreed to the published version of the manuscript.

**Funding:** This research received no external funding.

**Institutional Review Board Statement:** The study was conducted according to the guidelines of the Declaration of Helsinki, and approved by the local Ethics Committee (N. Registro CER Liguria: 135/2020-date 6 April 2020).

**Informed Consent Statement:** Informed consent was obtained from all subjects involved in the study.

**Data Availability Statement:** The datasets used and/or analyzed during the present study are available from the corresponding author on reasonable request.

**Conflicts of Interest:** The authors declare no conflict of interest.

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
