# Peer review of "Effects of Late Conversion from Twice-Daily to Once-Daily Slow Release Tacrolimus on the Insulin Resistance Indexes in Kidney Transplant Patients"

_2673-3943, doi:10.3390/transplantology2010005_

Round 1
Reviewer 1 Report
It is an interesting topic and of relevance to the follow-up of kidney transplant patients. Unfortunately, the number of patients is very small and the significance in retrospective analysis rather low.
An interventional approach by forming 2 groups and leaving one group on twice tacrolimus day (TAC BID) and switching the other to once a day (TAC OD) would have a greater significance. These groups should then be followed up for 8 months. This would be of much higher significance.
Anyway, the paper is well written and in this first retrospective Analysis TAC OD does not seem to be beneficial regarding glucose metabolism.
The lack of study design should be elaborated more in the discussion and suggestions for better power of future studies should be given.
There are further points which should be improved:
- Please define more clearly the in- and exlusion criteria in the methods part (also if combined pancreas/Kidney Transplantation was exclusion criterion
- Patients characteristis are inadequately explained. Please provide a table including mean time after Transplantation dialysis vintage, Diagnosis of DM at time of transplantation, etc., Living vs. deceased donation. Please keep in mind other transplant studies.
-
"Finally, we did not observe any association between Tac levels and blood levels of Glucose, Insulin, C-Peptide, HbA1c and uric acid." There was a significant difference in C-Peptide Levels. Please make this clearer.
-
Please report the exact nonsignificant P values and not only >0.05.
-
I would Combine Table 2 and 3
- Please revise the discussion section as recommended at the beginning.
Author Response
Dear Editor, Dear Reviewer,
We resubmit the paper entitled “Effects of late conversion from twice-daily to once-daily slow release tacrolimus on the insulin resistance indexes in kidney transplant patients” after revision according to the objections rose by the Reviewers.
Overall, we think that the criticisms were appropriate and constructive, and we hope that our new amended version of the manuscript could be considered suitable for publication.
In the revised text, changes are marked in red
REVIEWER 1
It is an interesting topic and of relevance to the follow-up of kidney transplant patients. Unfortunately, the number of patients is very small and the significance in retrospective analysis rather low.
An interventional approach by forming 2 groups and leaving one group on twice tacrolimus day (TAC BID) and switching the other to once a day (TAC OD) would have a greater significance. These groups should then be followed up for 8 months. This would be of much higher significance.
Anyway, the paper is well written and in this first retrospective Analysis TAC OD does not seem to be beneficial regarding glucose metabolism.
The lack of study design should be elaborated more in the discussion and suggestions for better power of future studies should be given.
There are further points which should be improved:
- Please define more clearly the in- and exclusion criteria in the methods part (also if combined pancreas/Kidney Transplantation was an exclusion criterion
As suggested, we explained in detail the inclusion/exclusion criteria in the Methods.
- Patient characteristics are inadequately explained. Please provide a table including the mean time after Transplantation dialysis vintage, Diagnosis of DM at time of transplantation, etc., Living vs. deceased donation. Please keep in mind other transplant studies.
As suggested, we added a table reporting the main patient characteristics (new table 1).
- "Finally, we did not observe any association between Tac levels and blood levels of Glucose, Insulin, C-Peptide, HbA1c, and uric acid." There was a significant difference in C-Peptide Levels. Please make this clearer.
Actually, C-peptide levels significantly decreased after the shift, but we did not find a correlation with Tac Levels (even if the correlation was almost significant, p=0.06). However, we added a supplemental table reporting the results of correlation analysis and some considerations on C-peptide (with proper references) in the Discussion.
- Please report the exact nonsignificant P values and not only >0.05.
- I would Combine Table 2 and 3
As suggested, we modified the tables and combined Table 2 and 3 (new table 2).
- Please revise the discussion section as recommended at the beginning.
As suggested, we revised the discussion, underlying the potential utility of a prospective interventional study (thank you for this suggestion).
Best regards,
Pasquale Esposito
Associate Professor of Nephrology,
Unit of Nephrology, Dialysis and Transplantation
Department of Internal Medicine, University of Genoa and Policlinico San Martino, Genoa, Italy
Reviewer 2 Report
The authors investigated the effects of the conversion from Tac-BID to Tac-OD on the basal insulin resistance indexes (HOMA and McAuley) and related metabolic parameters in a cohort of kidney transplant patients. The results showed that a significant decrease in Tac exposure was observed and no significant changes in Homa and McAuley indexes. Similarly, blood levels of Glucose, Insulin, HbA1c, lipids, and uric acid were unchanged between the two periods. The authors concluded that the conversion from Tac-BID to Tac-OD (reduced Tac exposure) has no significant effects on the basal insulin resistance indexes and related metabolic parameters in kidney transplant recipients. These results in this study may be useful for the conversion from Tac-BID to Tac-OD in kidney transplant patients. However, I think that there are some serious problems with this study. My specific comments are as follows.
1. The majority of patients (Italians?) are expected to be homozygous for the CYP3A5*3 allele.
Pharmacol Rep. 2010; 62: 1159-69.
Eur J Clin Pharmacol. 2011; 67: 47-54.
Many clinical studies showed that area under the concentration-time curve (AUC0-24) of Tac-BID and Tac-OD were equivalent on a mg for mg basis in stable kidney transplant recipients with the CYP3A5*3/*3 genotype (nonexpressers). On the other hand, in comparison with the Tac-BID formulation, the Tac-OD formulation is associated with a reduction in trough concentration. Therefore, sequentially monitoring trough concentrations may not always give a good indication of overall drug exposure.
2. The number of patients in this study is very small. In addition, only the results by univariate analysis are presented. The authors should carry out sample size calculation and power analysis. I think that the relationships between Tac preparations and basal insulin resistance indexes or related metabolic parameters should be examined by using such as covariance analysis after recruiting sufficient numbers of patients.
3. It is unknown when Tac-BID was converted to Tac-OD after 1-year post transplantation. That is to say, the duration of Tac-BID exposure in each patient is unknown.
4. As cited references [28, 29], similar studies have already been conducted in this research field, in this study, a new insight is not gained in light of these results.
Author Response
Dear Editor, Dear Reviewer,
We resubmit the paper entitled “Effects of late conversion from twice-daily to once-daily slow release tacrolimus on the insulin resistance indexes in kidney transplant patients” after revision according to the objections rose by the Reviewers.
Overall, we think that the criticisms were appropriate and constructive, and we hope that our new amended version of the manuscript could be considered suitable for publication.
In the revised text, changes are marked in red.
REVIEWER 2
The authors investigated the effects of the conversion from Tac-BID to Tac-OD on the basal insulin resistance indexes (HOMA and McAuley) and related metabolic parameters in a cohort of kidney transplant patients. The results showed that a significant decrease in Tac exposure was observed and no significant changes in Homa and McAuley indexes. Similarly, blood levels of Glucose, Insulin, HbA1c, lipids, and uric acid were unchanged between the two periods. The authors concluded that the conversion from Tac-BID to Tac-OD (reduced Tac exposure) has no significant effects on the basal insulin resistance indexes and related metabolic parameters in kidney transplant recipients. These results in this study may be useful for the conversion from Tac-BID to Tac-OD in kidney transplant patients. However, I think that there are some serious problems with this study. My specific comments are as follows.
1, The majority of patients (Italians?) are expected to be homozygous for the CYP3A5*3 allele.
Pharmacol Rep. 2010; 62: 1159-69.
Eur J Clin Pharmacol. 2011; 67: 47-54.
Many clinical studies showed that area under the concentration-time curve (AUC0-24) of Tac-BID and Tac-OD were equivalent on a mg for mg basis in stable kidney transplant recipients with the CYP3A5*3/*3 genotype (nonexpressers). On the other hand, in comparison with the Tac-BID formulation, the Tac-OD formulation is associated with a reduction in trough concentration. Therefore, sequentially monitoring trough concentrations may not always give a good indication of overall drug exposure.
This is an interesting observation that offers a new point of view in the study of the metabolic effects of immunosuppressive drugs. Actually, we have no data on the profile of genes involved in the pharmacokinetics of Tac, which evaluation is beyond the aim of this study. However, we added this consideration in the Discussion, hoping that it could be useful for future studies.
2. The number of patients in this study is very small. In addition, only the results by univariate analysis are presented. The authors should carry out sample size calculation and power analysis. I think that the relationships between Tac preparations and basal insulin resistance indexes or related metabolic parameters should be examined by using such as covariance analysis after recruiting sufficient numbers of patients.
We agree with the reviewer that the small number of patients evaluated is one of the main limitations of this study. However, we have not the possibility to enlarge the study. So, we added this consideration in the Discussion, calling attention to the necessity to perform larger and prospective studies on the same topic.
3. It is unknown when Tac-BID was converted to Tac-OD after 1-year post transplantation. That is to say, the duration of Tac-BID exposure in each patient is unknown.
Notably, since each patient commenced TacBID treatment immediately after the transplant, transplant age at the conversion corresponded to the duration of exposure to TacBID. Then, due to the small number of patients, we did not perform further subanalyses. We added this information in the Result section.
4. As cited in references [28, 29], similar studies have already been conducted in this research field, in this study, a new insight is not gained in light of these results.
As we reported, similar studies on the same topic have been already published, We think that the relative novelty of our study could be constituted by the fact that we evaluated the metabolic effects of Tac conversion for a longer time compared with previous studies. However, it should be also admitted that our results, such as those of previous studies, are not conclusive and many questions remain to explore (such as the effects of genetic polymorphisms on Tac pharmacokinetics, as you suggested). We added some considerations on these issues in the Discussion.
Reviewer 3 Report
General Comments
The authors retrospectively compared the level of basal insulin sensitivity indexes including HOMA and McAuley before and after the late conversion from TacBID to TacOD in 20 kidney transplant patients. The blood test was performed 6 months before to 8 months after the conversion which was 3.2 years after the kidney transplantation. And they found no significant effects on basal insulin sensitivity indexes and metabolic parameters in the present cohort except for the decreasing of the C-peptide level. The same kind of retrospective evaluations were already published in 3-4 studies previously so novelty is low. However, the patient number of these previous studies were also around 20-50.
Major revision
- The present study should show the detailed data of diabetes status in the inclusion and exclusion criteria before transplantation including the cause of kidney failure i.e. diabetes nephropathy. If the authors intended the study should be performed in non-diabetic patients, the authors should show the strict inclusion criteria.
- In line 165, the authors stated that they did not observe any association between Tac levels and blood levels of Glucose, Insulin, C-peptide, HbA1c, and uric acid. However, no data was available in the manuscript. If the authors would like to state this result, the authors need to show even in the supplementary data.
- The explanation of the reason why the authors did not emphasize the results of decreased C-peptide level in the TacOD was logically weak. The reviewer thinks the result of C-peptide should be more emphasized.
Author Response
Dear Editor, Dear Reviewer,
We resubmit the paper entitled “Effects of late conversion from twice-daily to once-daily slow release tacrolimus on the insulin resistance indexes in kidney transplant patients” after revision according to the objections rose by the Reviewers.
Overall, we think that the criticisms were appropriate and constructive, and we hope that our new amended version of the manuscript could be considered suitable for publication.
In the revised text, changes are marked in red.
REVIEWER 3
The authors retrospectively compared the level of basal insulin sensitivity indexes including HOMA and McAuley before and after the late conversion from TacBID to TacOD in 20 kidney transplant patients. The blood test was performed 6 months before to 8 months after the conversion which was 3.2 years after the kidney transplantation. And they found no significant effects on basal insulin sensitivity indexes and metabolic parameters in the present cohort except for the decreasing of the C-peptide level. The same kind of retrospective evaluations was already published in 3-4 studies previously so the novelty is low. However, the patient number of these previous studies were also around 20-50.
Major revision
1. The present study should show the detailed data of diabetes status in the inclusion and exclusion criteria before transplantation including the cause of kidney failure i.e. diabetes nephropathy. If the authors intended the study should be performed in non-diabetic patients, the authors should show the strict inclusion criteria.
As suggested, we explained in detail the inclusion/exclusion criteria in the Methods.
2. In line 165, the authors stated that they did not observe any association between Tac levels and blood levels of Glucose, Insulin, C-peptide, HbA1c, and uric acid. However, no data was available in the manuscript. If the authors would like to state this result, the authors need to show even in the supplementary data.
As suggested, we added a table (supplemental Table 1) reporting the results of correlation analysis.
3. The explanation of the reason why the authors did not emphasize the results of decreased C-peptide level in the TacOD was logically weak. The reviewer thinks the result of C-peptide should be more emphasized.
As suggested, we added some considerations (with proper references) in the Discussion about the potential meaning of modifications in C-peptide levels.
Best regards,
Pasquale Esposito
Associate Professor of Nephrology,
Unit of Nephrology, Dialysis and Transplantation,
Department of Internal Medicine, University of Genoa and Policlinico San Martino, Genoa, Italy
Round 2
Reviewer 1 Report
The reviewer's comments have been implemented. However, the significance of the study is low due to the retrospective design. However, this is now discussed in the discussion section.
The paper can therefore now be accepted at Transplantology.
Reviewer 2 Report
The manuscript has been revised well.
ï½¥ Please add the information of patients' race.
I have no further recommendations.
Reviewer 3 Report
I think all the points the reviewer suggested were appropriately corrected.